# Corrosion Testing of CrN_x_-Coated 310 H Stainless Steel under Simulated Supercritical Water Conditions

**DOI:** 10.3390/ma15165489

**Published:** 2022-08-10

**Authors:** Aurelia Elena Tudose, Florentina Golgovici, Alexandru Anghel, Manuela Fulger, Ioana Demetrescu

**Affiliations:** 1Institute for Nuclear Research Pitesti, POB 78, Campului Street, No. 1, 115400 Mioveni, Romania; 2Department of General Chemistry, University Politechnica of Bucharest, Splaiul Independentei Street, No. 313, 060042 Bucharest, Romania or or; 3National Institute for Laser, Plasma and Radiation Physics, Atomistilor Street, 077126 Magurele, Romania; 4Academy of Romanian Scientists, 3 Ilfov, 050094 Bucharest, Romania

**Keywords:** corrosion/oxidation, CrN_x_ coating, SEM, XRD, EIS

## Abstract

The paper’s aim is the assessment of corrosion behaviour of a CrN_x_-coated 310 H stainless steel under simulated supercritical water conditions (550 °C and 25 MPa) for up to 2160 h. The CrN_x_ coating was obtained by the thermionic vacuum arc (TVA) method. The oxides grown on this coating were characterized using metallographic and gravimetric analysis, SEM with EDS, and grazing incidence X-ray diffraction (GIXRD). A diffusion mechanism drives oxidation kinetics because it follows a parabolic law. By XRD analysis, the presence of Cr_2_O_3_ and Fe_3_O_4_ on the surface of the autoclaved CrN_x_-coated 310 H samples were highlighted. Corrosion susceptibility assessment was performed by electrochemical impedance spectroscopy (EIS) and linear potentiodynamic polarization. EIS impedance spectra show the presence of two capacitive semicircles in the Nyquist diagram, highlighting both the presence of the CrN_x_ coating and the oxide film formed during autoclaving on the 310 H stainless steel. Very low corrosion rates, with values up to 11 nm × year^−1^, obtained in the case of autoclaved for 2160 h, CrN_x_-coated samples indicated that the oxides formed on these samples are protective and provide better corrosion resistance. The determination of micro hardness Vickers completed the above investigation.

## 1. Introduction

The supercritical water-cooled reactor (SCWR) is one of the most promising reactor designs for Gen IV nuclear reactors. This type of reactor operates above the thermodynamic critical point of water (374 °C, 22.1 MPa). When supercritical water (SCW) is used as the coolant, no coolant recirculation pumps, steam generators, pressurizers, separators, or dryers are required, making the system more compact and simpler [1]. Due to the obvious unique physical and thermal features of supercritical water (SCW), supercritical water reactors are projected to have better thermal efficiency [2,3]. The SCWR’s design fulfils the criteria of sustainability, safety, and economics considered in the Generation IV International Forum [4].

However, due to the aggressiveness of the supercritical water, a major problem that arose was the corrosion of the materials used. As a result, a detailed investigation of the anticorrosive properties of candidate materials is required for the safe use of these nuclear reactor systems [5]. One of the most difficult aspects of developing such a reactor is selecting appropriate materials for internal components. Low swelling austenitic steels and ferritic-martensitic steels are promising materials for fuel assembly as well as for vessel internals components exposed to high neutron doses of 100 dpa [6], while for low dose components, nickel-based alloys, as well as high-strength austenitic steels are the candidate materials. At present, due to the good corrosion resistance in conventional nuclear reactors, austenitic stainless steels, ferritic-martensitic steels, and Ni-base alloys are considered promising candidate materials for SCWRs [5].

Austenitic stainless steels (ASS) are a class of material suitable to be used in supercritical water-cooled reactors (SCWRs) because of their great corrosion resistance, creep, and good radiation performance. Although they have a high corrosion resistance, when these types of steels are exposed to water under aggressive conditions (550 °C, 25 MPa) they are prone to stress corrosion cracking (SCC) [7,8,9].

The 310 H stainless steel (SS) is a promising material for the construction of the internal components of an SCWR. Because of the high chromium and nickel concentration, this austenitic alloy has considerable corrosion resistance. The development of a protective oxide on the alloy surface is promoted by chromium, while the existence of nickel in the composition of the alloy increases the stability of this oxide, particularly when it is exposed to hot water [5]. Although it has good resistance to corrosion, when exposed to high temperatures between 450 °C and 850 °C, this type of steel is sensitized and becomes susceptible to intergranular corrosion, due to chromium carbide precipitation at the grain boundaries [10,11] and chromium depletion near the grain boundaries [12]. Additionally, the high chromium content in 310 H makes this steel more susceptible to sigma phase precipitation, increasing the hardness [13,14]. Previous papers [15,16] have shown that chromium carbides and the precipitation of the sigma phase led to sensitization of the steel, making it more susceptible to intergranular corrosion when exposed to aggressive environments [17].

Therefore, one way to improve the corrosion resistance of the existing materials is the application on the surface of metallic or ceramic layers by various deposition techniques. There are several coating techniques: thermal spray (TS), chemical vapor deposition (CVD), physical vapor deposition (PVD), electrodeposition, sol-gel, pack cementation, cold spray, and hot dipping [18,19]. To obtain the desired characteristics of the coatings, the deposition parameters can be modified or, in some cases, can be applied with two or more coating techniques. Compared to other deposition techniques, the PVD technique has the benefit of lowering the deposition temperature below 500 °C. In this manner, it is possible to avoid undesirable diffusion processes or reactions between the substrate and the coating. The PVD method produces homogenous coatings with no pores or cracks. The coatings deposited using PVD are very dense, providing a thick enough layer. These types of coatings reduce the quantity of moisture or gas that may pass through the film. As a result, this kind of coating would be perfect for preventing material corrosion in a nuclear environment. This applies to current boiling water reactors (BWR), pressurized water reactors (PWR), and CANDU reactors, but also to Generation IV type reactors such as liquid metal-cooled reactors or supercritical water-cooled reactors (SCWR) [20,21].

Over time, in terms of enhancing corrosion resistance, several materials were deposited on the stainless steel surface: CrN, TiN, TiAlN, AlCrN, ZrO_2_, TiSiN, Ni_5_Al, Ni_50_Cr, Ni_20_Cr_5_Al, and Ni_20_Cr. Following tests performed in water at high pressures and temperatures, it was found that CrN and NiCrAlY-based coatings have a high resistance to oxidation at high temperatures and are thus promising candidate materials to form a corrosion-resistant coating on the fuel cladding. CrN_x_-based coating also protects the internal components of the SCWR from corrosion [22].

Chromium nitride (CrN_x_) coatings have been developed since the 1980s and early 1990s due to the need to obtain coatings with a higher hardness than TiN [23]. CrN_x_ is a strong and corrosion-resistant compound that has uses in a variety of areas, including medical implants [24], silver lustre ornamental coatings [25], and wear-resistant coatings for cutting tools, particularly where temperature corrosion resistance is required [26,27,28,29,30]. Chromium nitride is an interstitial compound in which nitrogen atoms reside in the octahedral spaces between the chromium atoms in an fcc lattice (i.e., NaCl), making the compound prone to stoichiometry deficiency. Moreover, in CrN_x_ films, as a secondary phase, a second interstitial compound, hexagonal Cr_2_N, can be produced [31,32]. Studies on the thermoelectric properties have shown that CrN_x_ has a low thermal conductivity, moderate electrical resistance, and a high Seebeck coefficient value. CrN_x_ has excellent mechanical properties (high hardness, low coefficient of friction, wear resistance) and good chemical properties (corrosion/oxidation resistance) [33,34]. CrN_x_ is an excellent option in high-temperature situations because of its enhanced temperature tolerance. [35,36,37].

CrN_x_ coatings can be obtained both by physical vapor deposition (PVD) techniques (reactive sputter deposition, plasma beam sputtering, cathodic arc evaporation, electron beam PVD, plasma TVA) and magnetron sputtering process, as well as chemical vapor deposition (CVD) techniques such as Plasma Enhanced Chemical Vapor Deposition (PECVD) [23,38].

The mechanical characteristics, phase compositions, and microstructure of CrN_x_ coatings may change depending on the deposition procedures and parameters used. Typically, CrN_x_ coatings have a columnar type of structure [23]. The phase structure of chromium changes as the nitrogen pressure increases, from the body-centred cubic lattice to a combination of Cr and hexagonal Cr_2_N to pure Cr_2_N, and then to a mixture of Cr_2_N and cubic CrN to pure CrN. These phases have varied physical characteristics due to their diverse chemical compositions and crystal shapes [36,39].

Because the performance and the stability of a material used for a SCW reactor depend on the corrosion behaviour and knowing that the corrosion process itself is an electrochemical process, electrochemical techniques are the best way to determine the reaction mechanisms. The main advantages of electrochemical techniques are their ability to detect extremely low corrosion rates, take real time measurements, and yield information at the reaction site, which can lead to well-established theoretical understanding.

Despite the advantages they can bring, only a few authors have used these modern electrochemical techniques to characterize metallic materials coated with chromium nitride layers used under special conditions [40,41,42], and extremely few [43,44] have studied the behaviour of the coating and the oxides formed on their surface.

This paper presents as a novelty the synthesis of chromium nitride coating on 310 H stainless steel substrate using the Thermionic Vacuum Arc technique, followed by testing under simulated supercritical water conditions (550 °C, 25 MPa) for up to 2160 h. Before and after exposure to deaerated static supercritical conditions, the surface coating performances were assessed using metallographic and gravimetric analysis, scanning electron microscopy (SEM) with an energy dispersive spectra detector (EDS), and grazing incidence X-ray diffraction (GIXRD). Another novelty is the use of modern electrochemical techniques such as electrochemical impedance spectroscopy (EIS) and linear potentiodynamic polarization to estimate in a short time the corrosion susceptibility of these CrN_x_-coated 310 H stainless steels when used as SCWR materials.

## 2. Materials and Methods

### 2.1. Coating Material

The substrate used in this work is a commercial 310 H stainless steel (SS) plate acquired from Outokumpu Stainless AB Company (Degerfors, Sweden). The plate, with a 2 mm thickness, was hot rolled and heat-treated at 1100 °C, followed by water quenching. The EDS analysis showed the following composition of the alloy used, presented in Table 1.

Rectangular samples of 25 mm × 15 mm (length × width) were cut from a sheet of 310 H SS. The samples were provided with a hole of 3 mm diameter at one end for mounting on the autoclave holder. Then, the specimens were mechanically sanded on several granulations of abrasive paper (#120, #240, #400, #600 m). Before deposition, the substrates were ultrasonicated in an isopropyl alcohol bath for 15 min, followed by blowing nitrogen under high pressure.

In this paper, the CrN_x_ thin films were obtained by injecting N_2_ gas (gaseous precursor) into a plasma of Cr (solid precursor) obtained by the TVA method. For coating, 1–3 mm Cr pellets of 99.99% purity purchased from NEYCO company (Vanves, France) and N_2_ gas of 99.9999% purity purchased from SIAD company (Bergamo, Italy) were used.

The selection of this coating was based on good mechanical properties (wear resistance, lower friction coefficient, higher toughness, high hardness) and good corrosion resistance (up to 700 °C) [33,34,35,36,37], which make them promising candidates as protective coatings for high temperature and high-pressure work.

### 2.2. Coating Method

The CrN_x_ coatings investigated in this research were deposited using the Thermionic Vacuum Arc (TVA) technique. Using TVA plasma, very thin, adherent, compact, smooth, and pure films in high vacuum conditions (of the order of 10^−5^ ÷ 10^−6^ mbar) were obtained. The characteristics of the plasma source are presented in previous works [45,46]. The used equipment has been developed at the Low Temperature Plasma Laboratory from National Institute for Laser, Plasma, and Radiation Physics (NILPRP, Magurele, Romania). The experimental arrangement of the TVA technique consists of a two electrodes system: the cathode is represented by an electron emitting heated tungsten filament, while the anode is a crucible containing the coating material, meaning Cr pellets, to be evaporated. To obtain CrN_x_ thin films, the N_2_ gas was injected into a Cr plasma obtained by the TVA method. The experimental setup is presented in Figure 1. To obtain uniform coatings on all sample surfaces, a rotating sample holder was used. The distance between the samples and the plasma source was 270 mm.

The basic principle is the ignition of an arc plasma in the vapors of the material of interest [45]. After ignition of chromium plasma, presented in detail in a previous paper [47], the N_2_ gas is laterally injected at a flow rate of 25 sccm (standard cubic centimetres per minute) with a high-precision flowmeter model MCV 0–50 sccm. The pressure in the reaction chamber during deposition was 2.5 × 10^−4^ mbar. Once the steady state Cr + N_2_ plasma is reached, the deposition was started. The electrical parameters used for deposition were filament current (If) of 55 A, discharge current (I_d_) of 3.4 A, and discharge voltage (U_d_) of 100 V. The duration of each deposition charge was 80 min. The thickness of the deposited CrN_x_ layer was estimated to be around 300 nm, based on gravimetric measurements using an analytical quartz balance.

### 2.3. CrN_x_ Coating Behaviour at High Temperature

At high temperatures, CrN_x_ reacts with O_2_ and forms Cr_2_O_3_, according to Equations (1) and (2).
(1)$$\frac{4}{3}\mathrm{CrN}+O_{2}=\frac{2}{3}{\mathrm{Cr}}_{2}O_{3}+\frac{2}{3}N_{2}$$
(2)$$\frac{2}{3}{\mathrm{Cr}}_{2}N+O_{2}=\frac{2}{3}{\mathrm{Cr}}_{2}O_{3}+\frac{1}{3}N_{2}$$

The transformation of nitride to oxide is accompanied by the formation of molecular nitrogen. At lower temperatures, there is an amount of nitrogen inside the oxide layer. At higher temperatures, molecular nitrogen is completely released into the gas phase and cannot be detected inside the oxide layer. Under non-oxidative conditions, the CrN_x_ coating is thermodynamically stable below 1060 °C, whereas the Cr_2_N coating is stable above that temperature.

### 2.4. Testing Method

After CrN_x_ deposition on 310 H alloy by the TVA method, to investigate the influence of the environment on corrosion behaviour, the samples were subjected to supercritical water. The determination was carried out in a static one-liter autoclave in water at 550 °C and a pressure of 25 MPa to mimic the supercritical environment. The exposure period was up to 2160 h. Supercritical water has special properties, such as a low oxygen content (less than 2 ppm), a conductivity at room temperature of about 0.2 μS/cm, and a pH value of 6.2. Samples of 310 H SS coated with CrN_x_ were withdrawn from the autoclave after a different period for the measurement of gravimetric weight gain of the samples after washing and drying, as well as structural and morphological examination and electrochemical testing. After each inspection, the autoclave solution was replaced with a fresh solution. Weight gain due to oxidation was assessed using a balance with an accuracy of 1 × 10^−5^ g. The fluctuation of the mass ΔW (in mg) was computed using the original weight and those acquired after oxidation.

The corrosion rates in mg dm^−2^ day^−1^ were calculated according to Equation (3): (3)$$V_{corr}=m_{d}\times \frac{0.0365}{\rho }$$
where *m_d_* is the variation of mass per unit area, *S*, and exposure time, *t*, of the sample, and *ρ* is the density of 310 H SS, which is 7.98 g/cm^3^. To determine the oxide layer thickness, it was assumed that the chromium oxide (Cr_2_O_3_) is the main oxide formed on the coated 310 H SS surface when exposed to supercritical conditions. This oxide has a density of 5.22 g/cm^3^ [48].

#### 2.4.1. Morphological and Structural Surface Analysis

Before and after exposure to supercritical static water conditions, the surfaces of the CrN_x_-coated 310 H SS samples have been characterized morphologically and structurally using metallographic analysis and SEM/EDS.

The oxide thickness, grain structure, and Vickers microhardness were highlighted using the Olympus GX71M optical microscope (Olympus Corporation, Tokyo, Japan). To evaluate the oxide layer thickness, small pieces were cut from the samples and wrapped in copper foil, embedded in conductive cupric resin, and ground (P #4000). Electrolytic etching in 10% oxalic acid solution, 6 V, for 5 to 25 s was used to emphasize the metallographic structure of the coated samples. The Vickers microhardness (MHV_0.1_) was determined by an OPL tester in an automatic cycle.

Scanning electron microscopy (SEM) was used to investigate the surface morphology of the coating before and after oxidation. For these analyses, a HITACHI SU5000 field emission scanning electron microscope (Hitachi, Tokyo, Japan) equipped with an energy-dispersive X-ray analyser (EDS, Oxford Instruments, Oxford, UK) was used.

The composition and structure of all samples were investigated by the Grazing Incidence X-ray Diffraction (GIXRD) method using a X’Pert PRO MPD Diffractometer (PANalytical B. V., Netherlands, Almelo) in a θ–2θ geometry using CuKα radiation (l = 1.5406 Å) and operating at room temperature. The GIXRD patterns were made at a grazing angle of 5° in the 20–100° range. To set the experimental analysis conditions, the X’Pert Data Collector program was used. The identification of the phase was made by referring to the International Center for Diffraction Data—ICDD (PDF-4+) database.

#### 2.4.2. Electrochemical Tests

Corrosion susceptibility assessment was performed by two electrochemical techniques: electrochemical impedance spectroscopy (EIS) and linear potentiodynamic polarization.

All the electrochemical measurements were performed using an electrochemical system PARSTAT 2273 potentiostat/galvanostat (Princeton Applied Research, AMETEK, OakRidge, TN, USA). A classical three-electrode electrochemical cell consisting of a working electrode (the CrN_x_-coated 310 H SS sample), an auxiliary electrode (graphite rod), and a saturated calomel reference electrode (SCE) was used to perform electrochemical measurements. Room temperature (22 ± 2 °C) was used for electrochemical testing. A chemically inert solution with a pH = 7.7–7.8 (0.05 M boric acid with 0.001 M borax solution), which did not affect the oxide layers’ features, was used as an electrolyte.

The impedance spectra [49] were obtained at open circuit potential (OCP) in the 100 kHz to 100 mHz frequency range, with an ac amplitude of 10 mV. The experimental EIS results were simulated with equivalent electrical circuits using ZView 2.90c software (Scribner Associates Inc., Southern Pines, NC, USA).

The linear polarization method with a scan rate of 0.5 mV/ s, in a potential range between −0.250 V up to +1.0 V vs. OCP, was applied.

## 3. Results and Discussion

### 3.1. Oxidation Kinetics

For the CrN_x_-coated 310 H SS samples, gravimetric analysis was performed after different periods of oxidation at 550 °C and 25 MPa. Weight gain data are presented in Figure 2 as a function of exposure time for CrN_x_-coated 310 H SS samples in supercritical static water conditions. The plots are based on limited experimental data (up to 2880 h) [5].

As can be seen from Figure 2, all samples gained weight during oxidation (the mass gain is very small, between 2.722 mg/dm^2^ and 3.1064 mg/dm^2^). The oxidation process is observed to follow a rate law described by Equation (4):(4)$$\frac{\Delta m}{S}=k\times t^{n}$$
where $\frac{\Delta m}{S}$ is the oxide weight gain (mg/ dm^2^), k is the rate constant, t is the exposure time (h), and n is the exponent.

Fitting the data with Equation (4), the parabolic oxidation constants (k) were determined. The obtained k values and R-squared value of trend (R^2^) for plots are presented in Table 2. The R-squared values were calculated to determine the reliability of the parabolic trend. Analysing Figure 2, it was established that for CrN_x_-coated 310 H SS samples, oxidation kinetics follow a parabolic law (a value of 0.5553 for n exponent) [50]. The transport of metal ions and oxygen through an oxide controls oxidation in parabolic kinetics. Using Wagner’s theory of metal oxidation [5], the parabolic constant may be determined as a function of the inward oxygen and the outward chromium (or iron) diffusion.

The oxide thickness and corrosion rate were computed at various stages of oxidation based on the results obtained from the sample weighing. Figure 3 depicts the oxide thickness and corrosion rate behaviour in supercritical deaerated water. The global layer thicknesses were also determined by gravimetric analysis.

Figure 3 shows that, as the oxidation period rises, the oxide thickness grows, while the corrosion rate decreases. The low computed corrosion rates of 10^−4^ mm/year suggest that CrN_x_-coated austenitic alloy has excellent resistance to general corrosion in water at high temperatures. Comparing with our previous study, in which uncoated 310 H stainless steel was autoclaved under the same conditions for the same periods of time [51], it can be said that a chromium nitride coating slightly decreases the corrosion rate, and the oxide film thickness is slightly smaller. It can therefore be concluded that, in this situation, both CrN_x_ coating, and also the increase of the oxide layer on its surface, led to the improved corrosion performance of 310 H stainless steel-coated material under supercritical conditions.

### 3.2. Morphological and Structural Characterization

#### 3.2.1. Metallographic Analysis (Optical Microscopy)

Using metallographic analysis, the CrN_x_-coated 310 H SS samples before and after exposure to supercritical water for 720 h, 1440 h, and 2160 h in the supercritical environment (water at 550 °C and 25 MPa) were characterized microstructurally.

The metallographic structure of the coated 310 H stainless steel (SS) samples before and after exposure was emphasized by electrolytic etching, after which the grain size was measured with an optical microscope. The microstructure of the samples at different periods of exposure in supercritical conditions are shown in Figure 4.

The micrographs of CrN_x_-coated samples before and after oxidation reveal an austenitic structure with well-marked grain boundaries. After exposure to water at supercritical temperature, the average grain size (G) varies slightly. The average grain size number have been determined in accordance with ASTM E-112 [52] in a cross-section by a linear interception method. The as-received samples uncoated have an ASTM number of G = 7.5 [51]. For all coated and oxidized samples, the values of the G number were approximately G = 7.5 ÷ 8. The average grain diameter was approximately 26.7 (for unoxidized coated sample), 24.6 µm (for coated sample oxidized 720 h), and 22.5 µm (for coated samples oxidized for 1440 h and 2160 h). The mechanism for variation of average grain size after exposure to supercritical temperature is strongly dependent on oxidation film composition and exposure time [53].

Vickers microhardness was evaluated using an OPL tester in an automated cycle with a 0.1 Kg load. Each sample received an average of ten indents. The microhardness of the unoxidized CrN_x_-coated 310 H SS was 178 Kgf/mm^2^, which slowly decreased to 174 Kgf/mm^2^ for the CrN_x_-coated 310 H SS oxidized for 720 h and 1440 h. For the CrN_x_-coated 310 H SS oxidized for 2160 h, the microhardness was further elevated to 175 Kgf/mm^2^.

#### 3.2.2. Scanning Electron Microscopy (SEM) Analysis

The morphologies of the CrN_x_-coated 310 H SS samples before and after exposure in supercritical water were studied using SEM analysis. The SEM surface morphology of coated 310 H SS samples before and after different periods of exposure for 720 h, 1440 h, and 2160 h in water at 550 °C and 25 MPa are presented in Figure 5.

We can see from Figure 5a, relatively uniform and adherent CrN_x_ deposition layers on the substrate. Once the exposure period increases, the appearance of oxide crystals over the CrN_x_ coating can be observed (Figure 5b–d).

The elemental compositions of the surfaces of the unoxidized coated sample and of three oxidized CrN_x__-_coated samples, for three periods of time, based on EDS surface analysis, are presented in Table 3.

From Table 3, it can be seen that, for the coated sample, nitrogen (13.16%) and chromium (34.33%) have the highest concentrations, proving that a chromium nitride film was formed on the sample’s surface. With oxidation time increasing, the content of nitrogen decreases significantly (from 13.16% to 3.73%) while the chromium content shows small changes. The oxygen peak has been identified in all spectra of the oxidized coated samples. The presence of oxygen proves the formation of chromium (Cr_2_O_3_) and iron oxides (Fe_3_O_4_) on the surface of the analysed samples. Iron probably diffuses through the chromium oxide layer, forming Fe-oxide on top of it [30].

#### 3.2.3. Grazing Incidence X-ray Diffraction (GIXRD) Measurements

The chemical composition and the structure of the investigated samples were assessed using the Grazing Incidence X-ray Diffraction (GIXRD) method. Figure 6 illustrates the GIXRD patterns of the CrN_x_-coated 310 H SS samples before and after exposure in supercritical water for 720 h, 1440 h, and 2160 h. As we can observe, TVA CrN plasma-coated 310 H stainless steel substrate (black line) revealed the presence of CrN with a 200 preferred orientation according to PDF Card No. 01-076-2494. For the coated samples analysed after different exposure periods in supercritical water, the XRD diffractogram presented in Figure 6 shows the presence of two types of oxides: Cr_2_O_3_ and Fe_3_O_4_.

The Cr_2_O_3_ (PDF Card No. 01-001-1294) pattern shows several diffraction peaks, the most important of which are at 24.6°, 33.5°, 36.3°, 50.4°, 54.9°, and 65.2° 2 theta, corresponding to the (012), (104), (110), (024), (116), and (300). The Eskolite phase of the Cr_2_O_3_ structure, in which the peaks are attributed to the rhombohedral structure, is thus confirmed [54]. The presence of Fe_3_O_4_ (PDF Card No. 01-002-1035) is highlighted by the appearance of the main peaks at 2 theta 35.3°, 43.3°, 74°, 89.9°, and 94.4°, associated with the orientations (311), (400), (533), (731), and (800).

The XRD patterns from Figure 6 depict the changes caused by different autoclaving periods. Furthermore, an increase in oxide thickness can be predicted due to the peak intensities of oxides increasing with autoclaving time.

### 3.3. Electrochemical Testing

#### 3.3.1. Electrochemical Impedance Spectroscopy

Analysis of protective properties of CrN coating thin films deposited on the 310 H SS before and after oxidation in supercritical water has been carried out by the electrochemical impedance spectroscopy method. This approach may evaluate the performances of surface layers while not speeding up electrochemical reactions at the metal/solution interface [55]. Impedance spectra for unoxidized and oxidized coated samples by the TVA method, measured at open circuit potential, are presented in Figure 7.

According to Figure 7a, the Nyquist diagrams reveal that the CrN_x_-coated 310 H SS sample oxidized for 720 h shows a better corrosion behaviour compared to the coated samples oxidized for 1440 h and 2160 h. This is indicated by the higher values of the polarization resistance (the diameter in Nyquist diagrams is a measure of polarization resistance, Rp). For the samples subjected to oxidation for different periods of time, the appearance of a capacitive semicircle, with a small diameter at high frequencies, and another capacitive semicircle, with a large diameter at medium and low frequencies, was observed. The first is associated with the CrN coating, and the other is associated with the oxide layer formed on the surface of the CrN coating following the autoclaving of the samples.

From the Bode plots, Figure 7b, it can also be observed that, for the CrN_x_-coated 310 H SS samples subjected to oxidation, higher impedance values were obtained. The impedance, |Z|, is directly proportional to oxide resistance. The high impedance modulus values found imply that these samples are corrosion resistant. The Bode diagram (Figure 7b) shows that there are two maxima of the phase angle, one at high frequencies and one at low frequencies. In the case of the coated samples, the impedances measured at low frequencies correspond to the corrosion processes that take place at the substrate-coating interface, and the impedances measured at high frequencies correspond to the corrosion. From the Nyquist diagrams (Figure 7a) the processes take place at the oxide-solution interface [56,57,58,59].

The protective character of the oxide films formed in supercritical static conditions is additionally given by the higher values of maximum phase angle (approx. 79° for the sample oxidized 720 h, 82° for the sample oxidized 1440 h, and 84° for the sample oxidized 2160 h). Figure 7b shows that the Bode plots present a maximum phase angle value close to 90°, indicating that the oxide films have a capacitive behaviour. Generally, a phase angle value closer to 90° indicates capacitive properties and higher protectiveness of the oxide [60].

In order to obtain quantitative data, the impedances of the layers deposited on the surface of the stainless steel were modelled using equivalent circuits. All experimental electrochemical impedance spectroscopy data were fitted using the equivalent electrical circuit model from Figure 8.

All the parameters obtained by the fitting of the experimental data with the proposed equivalent electrical circuit (EEC) are presented in Table 4. As can be seen from this table, a good fit of the data to this model was obtained.

The Bode plot (Figure 7b) analysis indicates that the phase angles do not exceed the value of −90°, which means that the obtained films are not fully capacitive. In the literature [51,55,60,61], it is assumed that this behaviour is due to surface roughness, film porosity, or local inhomogeneities, which requires the introduction of a non-ideal capacitor in the circuit equivalent to the oxide layer, which is usually a distinctive element. The impedance of the distributive element is represented by the constant phase element (CPE).

It can be seen that it was not possible to use a capacitive element to model the capacity of the oxide layer. The deviation from an ideal capacitor can be observed following the values for CPE-T, which are lower than 1, so the oxide film has inhomogeneities or pores [51,56,61,62,63]. Additionally, the higher value of the charge transfer resistance obtained in the case of the coated and oxidized sample for 2160 h indicates that the deposited film has a more protective character and confers better corrosion resistance compared to the films formed on the surface of the coated and oxidized samples for a shorter period of time. The table also shows that a Chi-square value (χ2) ≈ 10^−4^ was found, so it can be said that the experimental data fit well with the proposed EEC because the fitting errors were quite small.

#### 3.3.2. Potentiodynamic Polarization Tests

To observe the CrN_x_-coated 310 H SS samples’ behaviour before and after exposure for different periods in supercritical water (550 °C, 25 MPa) the potentiodynamic plots were recorded and are presented in Figure 9.

As can be observed, the increase in autoclaving time leads to the shift of the corrosion potential to more electropositive values and the decrease of the corrosion current density value. All autoclaved samples have smaller values of corrosion current density, which means lower corrosion rates than the non-autoclaved sample.

Applying the Tafel slope extrapolation method and polarization resistance methods for the polarization plots from Figure 9, the electrochemical parameters shown in Table 5 were obtained. The main parameters are the corrosion rate (v_corr_), corrosion potential (E_corr_), current density (*i_corr_*), and polarization resistance (*R_p_*). Based on the electrochemical parameters obtained, the porosity, *P* (%), of the oxide films and the protective efficiency, *P_i_* (%), have been evaluated quantitatively using Equations (5) and (6), respectively [54]. The porosity is an important parameter for defect densities evaluation [64]. A low porosity indicates a favourable protective efficiency. Different electrochemical methods may be employed to assess porosity, the simplest being the polarization resistance approach [65,66].
(5)$$P_{i}\left(\%\right)=\left[1-\left(\frac{i_{corr}}{i_{corr}^{0}}\right)\right]\times 100$$
where *i_corr_* and $i_{corr}^{0}$ are the corrosion current density of the oxide film and substrate (CrN_x_-coated 310 H), respectively.
(6)$$P\left(\%\right)=\left(\frac{R_{p_{S}}}{R_{p_{coat}}}\right)\times 10^{-\left(\frac{\Delta E_{corr}}{\beta_{a}}\right)}$$
where *P* is the total oxide film porosity, *R_ps_* is the polarization resistance of the substrate (CrN_x_-coated 310 H SS), *R_pcoat_* is the polarization resistance of the oxidized-coated 310 H SS sample, ΔE_corr_ is the difference between the free corrosion potentials of the oxidized coated 310 H SS and the substrate, and *β_a_* is the anodic Tafel slope for the substrate.

From Table 5 and Figure 9, it can be seen that the CrN_x_-coated 310 stainless steels oxidized for a different period in water at supercritical static conditions provide better corrosion resistance compared to the CrN_x_ covered but non-oxidized sample. This is indicated by lower values of passivation current densities, more electropositive values of corrosion potentials, and lower values of corrosion rate.

It can be observed that, as the oxidation time increases, the corrosion current density decreases, the polarization resistance increases, and the corrosion rate decreases.

As is well known, the forms of corrosion that can occur in stainless steels are types of local corrosion such as pitting and intergranular corrosion. At the same time, some pits could be present with an intergranular form, which develop due to some corrosion of the grain being adjacent to grain boundaries. Often, as corrosion results, chromium-rich carbides are formed, as has been investigated in the literature [67].

In the conditions of a SCW reactor, hydrogen processes such as embrittlement can affect corrosion properties of alloys and have been studied intensively in relation to the influence on the mechanical properties as well.

It is well known that during oxidation of a metal at high temperatures, significant amounts of hydrogen gas can be released. In the case of stainless steel, the corrosion products formed are typically magnetite/hematite oxide layers. In the case of a nuclear reactor (particularly an SCWR), hydrogen is also a primary product of the radiolytic breakdown of water in the in-core region of the reactor.

The presence of excess hydrogen can lead to alloy failure in nuclear power systems, as hydrogen diffuses into the metal, causing stress corrosion cracking, hydrogen embrittlement, or delayed hydride cracking, which weakens the metal structure.

Some researchers [68,69] have demonstrated that the hydrogen evolved from the metal’s surfaces may be sufficient to reduce the accumulation of dissolved oxygen, thus inhibiting corrosion during the operation of the SCWR.

When comparing the values of the kinetic parameters obtained with those obtained for stainless steel samples subjected to autoclaving under the same conditions and for the same periods of time, in a study carried out in our previous paper [51], we observed that CrN coating by the TVA method leads to an improved corrosion performance of 310 H SS samples oxidized in supercritical static conditions.

Moreover, the protective efficiency and porosity obtained values reveal that the CrN_x_-coated 310 H SS samples oxidized in water at 550°C for 2160 h have the highest protective efficiency (85.06%) and the lowest porosity (0.1%). In concordance with the literature data [70,71,72], when increasing exposure time, protection efficiency increases and porosity decreases. Lower corrosion current densities and lower corrosion rates, as well as higher polarization potentials and higher polarization resistances, showed the best oxide/coating protective efficiencies, which means a higher ability to prevent corrosion.

## 4. Conclusions

In this study, CrN_x_ has been successfully deposited on 310 H stainless steel substrate using the Thermionic Vacuum Arc technique. This appears to be the first report related to CrN_x_-coated 310 H stainless steel autoclaved in supercritical static conditions for different periods, as well as the morphological, structural, and electrochemical characterization of these samples.

Based on the experimental data, we can present the following performances achieved after autoclaving for various periods of CrN_x_-coated 310 H stainless steel:Gravimetric analysis revealed that the CrN_x_-coated 310 H stainless steel oxidized for different periods of time at 550 °C and 25 MPa following parabolic rate, suggesting that oxidation kinetics is driven by a diffusion mechanism.The thickness of the oxides increased, while the corrosion rates decreased once the oxidation time increased. The low values of the computed corrosion rates suggested that CrN_x_-coated stainless steel has a good, generalized corrosion behaviour in supercritical static conditions.The microstructural analysis revealed an austenitic structure with polyhedral grains and well-defined grain boundaries.The presence of Cr_2_O_3_ and Fe_3_O_4_ on the surface of the autoclaved samples was highlighted by XRD analysis.The presence of two capacitive semicircles in the Nyquist diagram, as well as the second maximum of the phase angle in the Bode diagram, for CrN_x_-coated and autoclaved samples, reveals the presence on the sample surface of both the CrN layer and the oxide layer formed after autoclaving.Very low corrosion rates, obtained in the case of autoclaved CrN_x_-coated samples, indicated that the oxides formed on these samples are protective and provide better corrosion resistance.A good correlation between metallographic and gravimetric analysis, SEM, and electrochemical results was found.The low values of the corrosion rate for the CrN_x_-coated samples, compared to the uncoated ones, highlighted that this type of coating significantly improves the anticorrosive properties of 310 H stainless steel tested in supercritical conditions at 550 °C and 25 MPa.

## Figures and Tables

**Figure 1 materials-15-05489-f001:**
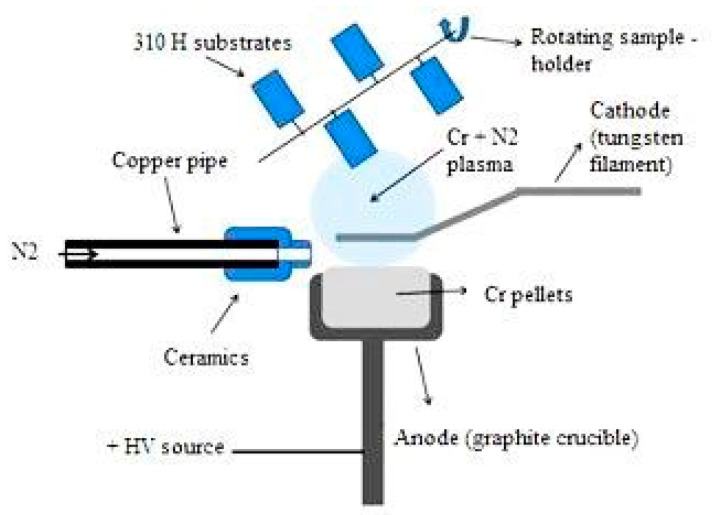
Experimental setup for TVA coating technique used to develop a CrN_x_ thin film on the 310 H SS surface.

**Figure 2 materials-15-05489-f002:**
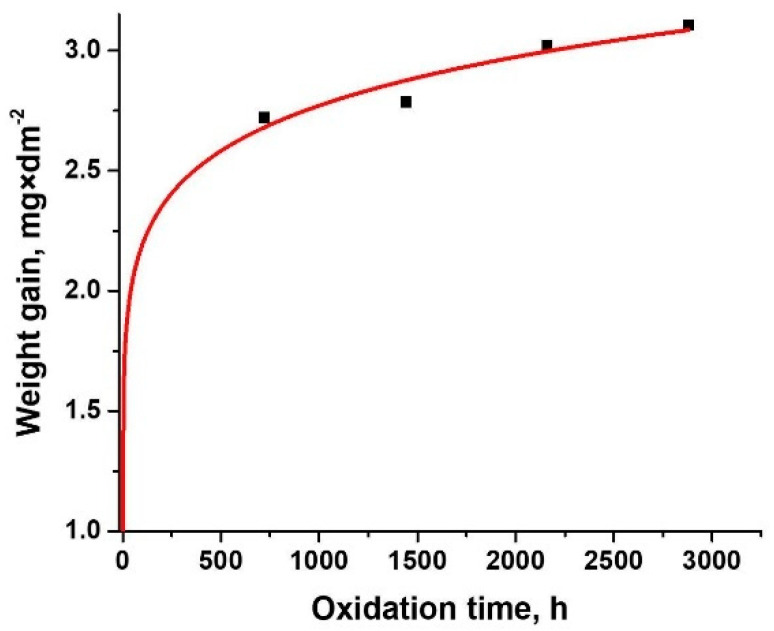
Oxidation kinetics of CrN_x_-coated 310 H SS samples after different periods of oxidation in supercritical water.

**Figure 3 materials-15-05489-f003:**
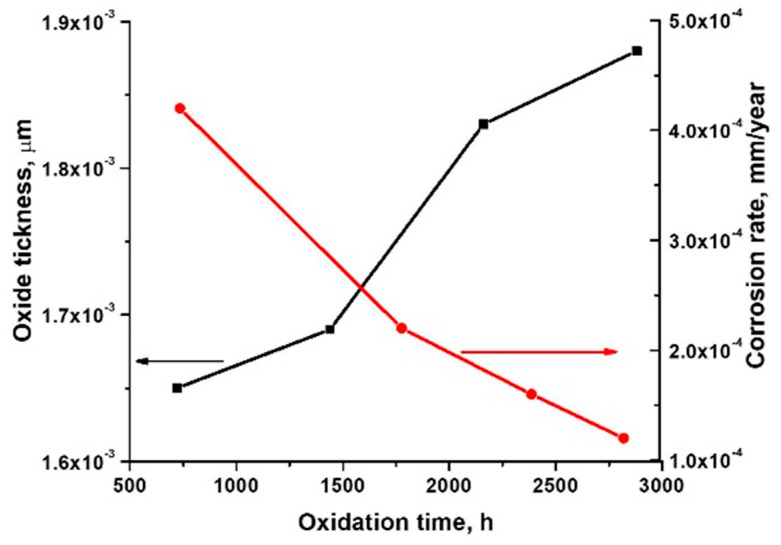
The behaviour of corrosion rate (red line) and oxide thickness (black line) depending on exposure period in supercritical static conditions of the CrN_x_-coated 310 H samples.

**Figure 4 materials-15-05489-f004:**
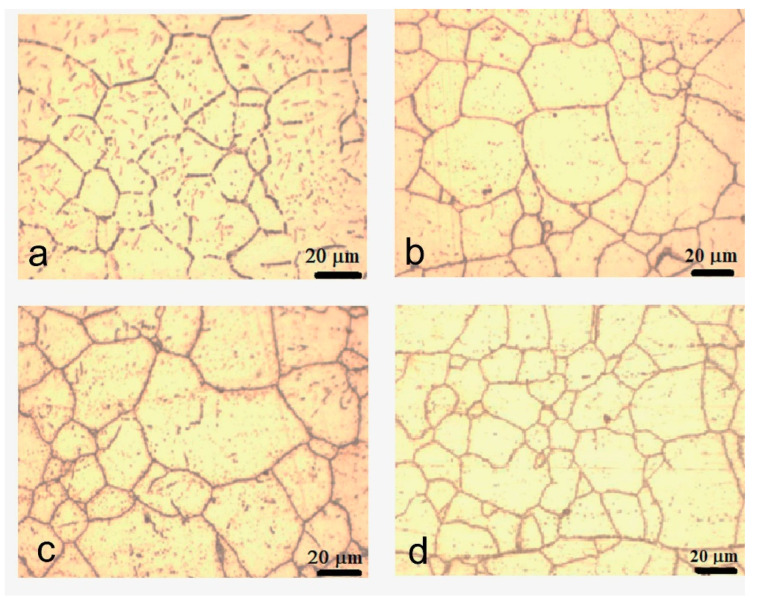
Microstructure for coated 310 H SS samples before (**a**) and after exposure in supercritical water for 720 h (**b**), 1440 h (**c**), and 2160 h (**d**) at 550 °C at magnifications of (500×).

**Figure 5 materials-15-05489-f005:**
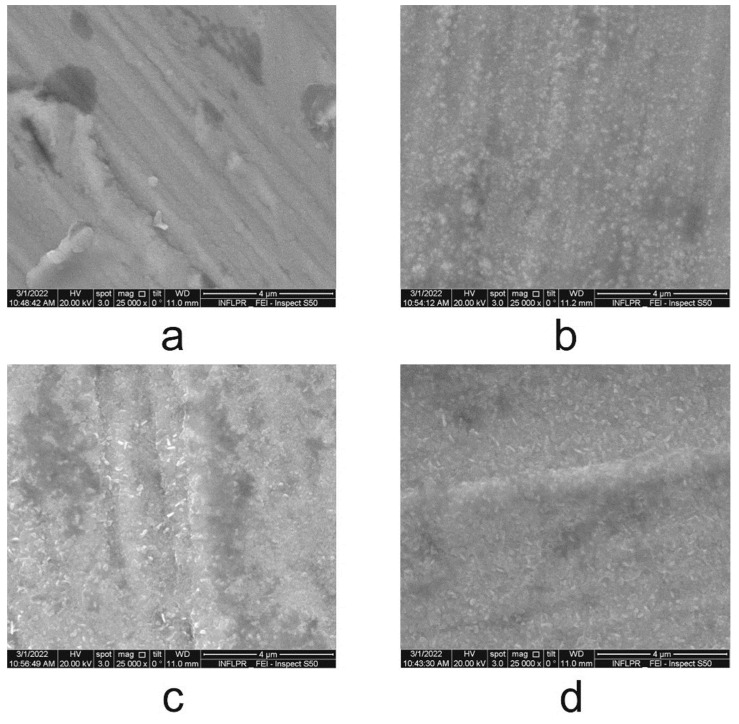
The surface morphologies of coated 310 H SS samples before and after different periods of exposure in water at 550 °C and 25 MPa at a magnification of 10,000×: (**a**) 0 h, (**b**) 720 h, (**c**) 1440 h, (**d**) 2160 h. Inset EDS analysis.

**Figure 6 materials-15-05489-f006:**
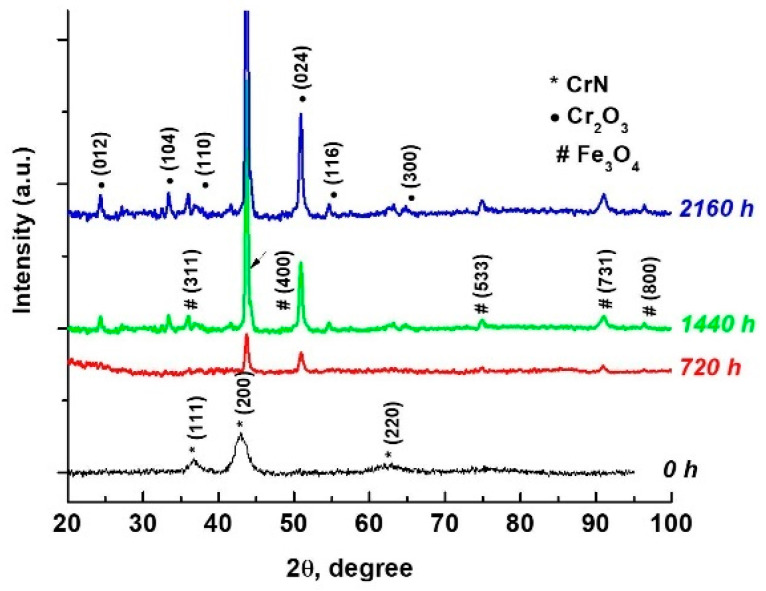
XRD diffractogram of coated 310 H SS samples before and after different periods of exposure in water at 550 °C and 25 MPa.

**Figure 7 materials-15-05489-f007:**
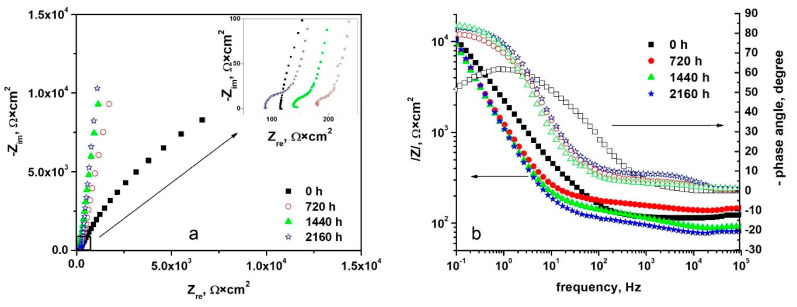
Nyquist (**a**) and Bode (**b**) diagrams for the CrN_x_-coated 310 H SS samples before and after oxidation in water at 550 °C and 25 MPa.

**Figure 8 materials-15-05489-f008:**
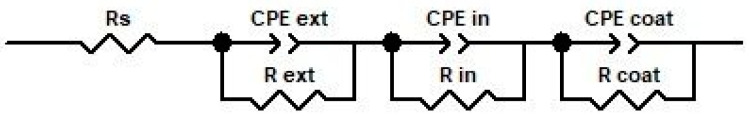
Electrical equivalent circuits proposed for fitting the experimental impedance spectra for the CrN_x_-coated 310 H SS samples.

**Figure 9 materials-15-05489-f009:**
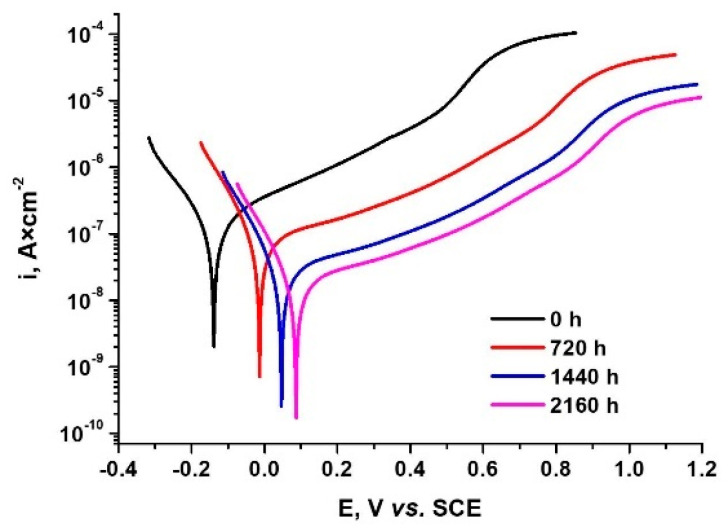
Potentiodynamic plots for CrN_x_-coated 310 H SS sample before and after exposure for different periods in supercritical water (550 °C, 25 MPa) at a scan rate of 0.5 mV × s^−1^.

**Table 1 materials-15-05489-t001:** Chemical composition of 310 H SS.

Elements, %wt.						
C	N	Si	Cr	Mn	Fe	Ni
0.063	0.04	0.71	24.13	1.61	54.34	19.03

**Table 2 materials-15-05489-t002:** Kinetic parameters for CrN_x_-coated 310 H SS samples in supercritical conditions.

Kinetic Equation	k	n	R^2^
y = 0.048 × t^0.555^	0.048	0.5553	0.9978

**Table 3 materials-15-05489-t003:** Chemical composition for the unoxidized and oxidized in supercritical water CrN_x_-coated sample.

Oxidation Time, h	%wt.							
	C	N	O	Si	Cr	Mn	Fe	Ni
0	0.66	13.16	-	1.06	34.33	1.63	36.76	12.41
720	0.18	4.27	15.16	1.16	33.43	1.94	32.65	11.19
1440	0.14	3.76	16.38	1.46	29.53	1.98	35.00	11.74
2160	0.26	3.73	14.53	1.16	29.10	2.39	36.61	12.22

**Table 4 materials-15-05489-t004:** The values of EEC elements for CrN_x_-coated 310 H SS samples before and after different periods of oxidation in water at 550 °C and 25 MPa.

Element		Oxidation Time, h		
	0	720	1440	2160
Rs, Ω × cm^2^	112.3	172.2	133.8	86.5
CPEext—T, F × cm^−2^		9.37 × 10^−5^	8.58 × 10^−6^	5.07 × 10^−5^
CPEext—P		0.634	0.891	0.705
IRext, Ω × cm^2^		37.03	22.13	23.39
CPEin—T, F·cm^−2^		2.13 × 10^−3^	5.64 × 10^−4^	2.43 × 10^−3^
CPEin—P		0.566	0.551	0.443
Rin, Ω × cm^2^		335.1	65.88	5807
CPEcoat—T, F × cm^−2^	1.15 × 10^−4^	1.66 × 10^−4^	1.65 × 10^−4^	1.566 × 10^−4^
CPEcoat—P	0.728	0.917	0.933	0.98
Rcoat, Ω × cm^2^	111.6	6.34 × 10^8^	7.86 × 10^6^	5.93 × 10^5^
Chi-squared (χ^2^)	1.4 × 10^−4^	1.7 × 10^−4^	1.9 × 10^−4^	1.8 × 10^−4^

**Table 5 materials-15-05489-t005:** Electrochemical parameters of CrN_x_-coated 310 H SS samples before and after exposure for different periods in supercritical water (550 °C, 25 MPa).

Oxidation Period, h	Tafel Slopes Method			Polarization Resistance Method		P_i_, %	P, %
	E_corr_, mV	i_corr_, nA × cm^−2^	V_corr_, μm × year ^−^^1^	R_p_, MΩ × cm^2^	i_corr_, nA × cm^−2^		
0	−140 ± 0.03	70.3 ± 0.02	0.7412 ± 0.02	0.37 ± 0.02	68.1 ± 0.02	–	–
720	−5 ± 0.01	32.1 ± 0.02	0.0338 ± 0.01	0.71 ± 0.03	31.2 ± 0.02	54.38 ± 0.01	2.33 ± 0.001
1440	+61 ± 0.01	16.5 ± 0.01	0.0174 ± 0.02	1.32 ± 0.03	15.5 ± 0.02	76.53 ± 0.01	0.27 ± 0.001
2160	+83 ± 0.01	10.5 ± 0.01	0.0111 ± 0.01	2.12 ± 0.04	10.4 ± 0.01	85.06 ± 0.01	0.10 ± 0.001

## Data Availability

The data presented in this study are available upon request.

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
