# Peer review of "Corrosion Testing of CrNx-Coated 310 H Stainless Steel under Simulated Supercritical Water Conditions"

_materials, 2022, doi:10.3390/ma15165489_

Round 1

Reviewer 1 Report

Dear authors :your work has major revision as commented bellow:

1- Abstract should rewrite and  numerical data must be added.

2-In introduction , the new references in 2015-2022 must be added.

3-SEM and EDX pictures are not clear. the EDX picture should divide and explain more about it.

4-The nyquist  diagram is not good , How to consider it?

5-grammatical and spelling errors should correct.

quist diagram a circul

Reviewer 2 Report

“...The selection of this coating was based on reported in other papers good mechanical 150 properties...” The article does not present references about these other works.

How thick is the CrNx coating?

In the figure 3, Is the black line chart  correspond to oxide tickness? Not explained in figure

Does the increase in oxidation thickness itself contribute to decreasing the corrosion rate?

What is the mechanism for increasing average grain size after exposure to water at supercritical temperature?

“...The oxygen  peak appeared in all spectra of the oxidized coated samples. This means that on all the oxidized coated samples a very thin oxide was formed. The peaks for Fe, Ni, Mn and Si  appear from the substrate...” This depends on the energy  beam for EDS analysis. At higher energies, the interaction volume is greater. Or, Fe3O4 phase formation , according to GIXRD

Reviewer 3 Report

The manuscript studies the corrosion testing results of CrNx coated 310 H SS under simulated supercritical water conditions. The submission needs a major revision by considering the following comments:

1- The quality of the figures is low and a high-quality version should be added. 

2- In the introduction, the authors should compare the techniques for electrochemical methods for corrosion studies like weight loss, TP, EIS, LPR, ... Why the current techniques were used?

3- The optical micrographs and SEMs in Fig. 4,5 are very poor in quality. Please replace them.

4- The manuscript needs some explanation using the references about the absorption of hydrogen from water which can affect the corrosion properties of an alloy. Please cite the following references: https://doi.org/10.1016/j.elecom.2021.107169 https://doi.org/10.1299/jsmeibaraki.2019.27.421

5- Please itemize the conclusions by numbers.

6- The abstract should be improved to represent the main finding of the manuscript.

Reviewer 4 Report

This paper presents the corrosion results of CrNx coated 310 H Stainless Steel under simulated supercritical water conditions. Some concerns are given below.

1. Quality of figures 1, 2, 5(inset EDS) and 8 have to be improved.

2. The base surface morphology (SEM images) of the samples before corrosion tests should be included.

3. The mechanisms of corrosion and the corrosion products should be explained in detail in the results and discussion.

4. Conclusions seem to be very qualitative. Addition of ballpark numbers would add some value. 

5. There are some minor English and formatting corrections required. Subscripts in the manuscript have to be checked. 

6. The authors should explain the novelty of this work in the introduction.

7. Why particularly CrNx coating? Why not other coatings?

8. Did the authors perform any tests to prove the adherence of the coating to stainless steel? How adherent is the coating?

Round 2

Reviewer 1 Report

Accept

Reviewer 2 Report

I accept corrections in the manuscript

Reviewer 3 Report

The revised version improved compared to the previous version. I recommend publication in the current status.